Effects of taphonomic deformation on geometric morphometric analysis of fossils: a study using the dicynodont Diictodon feliceps (Therapsida, Anomodontia)

http://orcid.org/0000-0002-0596-623X Kammerer Christian F. 1 christian.kammerer@naturalsciences.org
Deutsch Michol 2
http://orcid.org/0000-0001-8215-3796 Lungmus Jacqueline K. 2 3
Angielczyk Kenneth D. 2 3
1 North Carolina Museum of Natural Sciences , Raleigh, NC , USA
2 Field Museum of Natural History , Chicago, IL , USA
3 University of Chicago , Chicago, IL , USA
Young Mark
Electronic publication date: 2020 Oct 7
Publication date: 2020
Volume: 8
Electronic Location ID: e9925
Received 2020 Apr 27; Accepted 2020 Aug 21
Copyright: © 2020 Kammerer et al.
Copyright year: 2020
Copyright holder: Kammerer et al.
License: This is an open access article distributed under the terms of the Creative Commons Attribution License, which permits unrestricted use, distribution, reproduction and adaptation in any medium and for any purpose provided that it is properly attributed. For attribution, the original author(s), title, publication source (PeerJ) and either DOI or URL of the article must be cited.
License URL: https://creativecommons.org/licenses/by/4.0/

Keywords: Taphonomy, Morphometrics, Synapsida, Dicynodontia, Permian, Simulations

Funding: National Science Foundation of the United States of America NSF DEB 0608415 Deutsche Forschungsgemeinschaft DFG KA 4133/1-1 This work was supported by grants from the National Science Foundation of the United States of America (NSF DEB 0608415) and the Deutsche Forschungsgemeinschaft (DFG KA 4133/1-1) to Christian F. Kammerer. The funders had no role in study design, data collection and analysis, decision to publish, or preparation of the manuscript.

==============================
Taphonomic deformation, the distortion of fossils as a result of geological processes, poses problems for the use of geometric morphometrics in addressing paleobiological questions. Signal from biological variation, such as ontogenetic trends and sexual dimorphism, may be lost if variation from deformation is too high. Here, we investigate the effects of taphonomic deformation on geometric morphometric analyses of the abundant, well known Permian therapsid Diictodon feliceps. Distorted Diictodon crania can be categorized into seven typical styles of deformation: lateral compression, dorsoventral compression, anteroposterior compression, “saddle-shape” deformation (localized collapse at cranial mid-length), anterodorsal shear, anteroventral shear, and right/left shear. In simulated morphometric datasets incorporating known “biological” signals and subjected to uniform shear, deformation was typically the main source of variance but accurate “biological” information could be recovered in most cases. However, in empirical datasets, not only was deformation the dominant source of variance, but little structure associated with allometry and sexual dimorphism was apparent, suggesting that the more varied deformation styles suffered by actual fossils overprint biological variation. In a principal component analysis of all anomodont therapsids, deformed Diictodon specimens exhibit significant dispersion around the “true” position of this taxon in morphospace based on undistorted specimens. The overall variance associated with deformation for Anomodontia as a whole is minor, and the major axes of variation in the study sample show a strong phylogenetic signal instead. Although extremely problematic for studying variation in fossil taxa at lower taxonomic levels, the cumulative effects of deformation in this study are shown to be random, and inclusion of deformed specimens in higher-level analyses of morphological disparity are warranted. Mean morphologies of distorted specimens are found to approximate the morphology of undistorted specimens, so we recommend use of species-level means in higher-level analyses when possible.

Introduction

Geometric morphometrics is a well-established tool for addressing biological questions related to shape (Bookstein, 1991; Zelditch et al., 2004). The discriminatory power of geometric morphometrics allows for fine-scale resolution of shape differences between organisms or their parts. Although initially developed in the study of extant organisms, geometric morphometrics has also been used extensively to study fossil taxa, including representatives of all major vertebrate groups (Botha & Angielczyk, 2007; Deeming & Mayr, 2018; Pérez-Ben, Báez & Schoch, 2019; Price et al., 2019; Felice et al., 2019). These techniques have been used for biomechanical modeling (Pierce, Angielczyk & Rayfield, 2008; Polly et al., 2016) and to quantify the evolution of morphological disparity (Brusatte et al., 2012; Lungmus & Angielczyk, 2019), evolutionary rates (Adams, 2014), and ecological adaptions (Grossnickle & Newham, 2016). However, in contrast to extant systems, where the effects of biased sampling can be easily controlled, geometric morphometric studies on fossils include an additional, abiotic source of morphological variation: that of taphonomic deformation.

Taphonomic deformation includes all sources of postmortem shape change in a biological structure, but is mostly used to refer to distortion as a result of geological processes in the surrounding rock (Angielczyk & Sheets, 2007). Although taphonomic deformation of fossils can provide useful information in a geological context (Ramsay & Huber, 1983), in the field of paleobiology it is a problematic source of error, obscuring biologically important characteristics of fossil organisms. Various retrodeformation methods have been proposed to quantify and correct distortion in fossils (Wellman, 1962; Hughes & Jell, 1992; Rushton & Smith, 1993; Motani, 1997; Gunz et al., 2009; Arbour & Currie, 2012; Molnar et al., 2012; Tallman et al., 2014; Lautenschlager, 2016), but tests have shown that these techniques do not always recover underlying biological variance in specimens of known original morphology (Angielczyk & Sheets, 2007; Tschopp, Russo & Dzemski, 2013; Schlager et al., 2018). Given this problem, and the substantial practical difficulties involved in the retrodeformation of large sample sizes, a better understanding of the effects of taphonomic variation on biological variance in fossils is desirable.

For geometric morphometric analyses in particular, it is currently uncertain to what degree taphonomic deformation can “overwrite” the underlying signal of biological shape variation. Does distortion obscure variation intraspecifically, interspecifically, or even at higher clade levels? To address these questions, study systems comprising taxa represented by large sample sizes known from a variety of localities and stratigraphic intervals are needed. Although a frequent target for morphometric and other disparity-based analyses, few groups of terrestrial vertebrates satisfy these criteria. A noteworthy exception, however, can be found in the non-mammalian Synapsida.

Non-mammalian synapsids are one of the major groups of Paleozoic terrestrial vertebrates, and dominated ecosystems in both abundance and richness from the Pennsylvanian through the Middle Triassic (Kemp, 1982; Rubidge & Sidor, 2001; Angielczyk & Kammerer, 2018). Non-mammalian synapsids have a rich fossil record, especially in the biostratigraphically well-resolved Beaufort Group of South Africa, which preserves a nearly continuous record of vertebrate fossils from the middle Permian through the Middle Triassic (Rubidge, 1995, 2005; Smith, Rubidge & Van der Walt, 2012; Rubidge et al., 2016). The period of time in which non-mammalian synapsids were dominant encompasses several major biotic events in Earth history, such as the origin of modern terrestrial trophic structure (Olson, 1966) and the Permo-Triassic mass extinction (Ward, Montgomery & Smith, 2000; Ward et al., 2005; Smith & Botha-Brink, 2014). Given the plentiful available specimens, heavily sampled faunal assemblages, lengthy evolutionary history, and wide variety of inferred ecologies (Kemp, 1982) in non-mammalian synapsids, this group represents an ideal system in which to study morphospace occupation and disparity over time (Kammerer, 2009).

Although a number of synapsid species are represented by large sample sizes ranging into the hundreds of specimens (Smith, Rubidge & Van der Walt, 2012; Codron et al., 2017), the group also includes numerous stratigraphically and phylogenetically important species represented by very few individuals (e.g., all but one species in the therapsid subclade Burnetiamorpha are known from a single skull; Sidor, 2015; Day et al., 2018). Ideally, analyses of synapsid morphological disparity should include both abundant taxa as well as rare species. Of concern, however, are the possible effects deformation may have on morphospace occupation across taxa with varying sample sizes. A singleton taxon known only from a highly distorted specimen will to some degree be displaced from its “true” position in morphospace, with the associated degree of error unknowable pending discovery of new, undistorted specimens (though it can be roughly estimated using retrodeformation in some cases). The likelihood of preserving undistorted specimens is higher in an abundant taxon, which can also provide the means to quantify the amount of error incurred through the inclusion of distorted specimens in morphometric analyses. However, samples of an abundant fossil taxon carry their own problems. Abundant taxa, especially those with wide geographic and stratigraphic ranges, are more likely to exhibit a broader range of styles of deformation than specimens of a rare one. A distorted singleton can only be displaced from its “true” position in morphospace unidirectionally, but numerous specimens distorted in different ways may disperse from their “true” position in a variety of directions to varying degrees (Angielczyk & Sheets, 2007; Hedrick et al., 2019). In a specimen-level morphometric analysis, increased dispersion of distorted individuals around their “true” position in morphospace may artificially inflate some disparity metrics (e.g., total morphospace occupation of a taxon/clade). Alternatively (and possibly concurrently), a widely dispersed cluster of distorted specimens of an abundant taxon relative to rare ones may result in the underestimation of distance-based measures of disparity (with taxon-to-taxon distance in morphospace artificially reduced by distorted specimens of one taxon impinging on the “true” morphospace of another). This issue can also confound use of morphometric analysis for taxon discrimination, if specimens of a rare but biologically distinct taxon fall within the inflated specimen cloud of an abundant taxon due to distortion in the latter. A final, most distressing possibility is that high levels of deformation in abundant taxa completely overwhelm underlying biological variance in the study group, resulting in a morphospace that summarizes abiotic distinctions between individuals rather than any biologically meaningful variables.

Given these potential difficulties, it is important to address certain questions regarding taphonomic deformation in morphometric studies, focusing here on the synapsid fossil record as a case study. For example, is deformation of synapsid specimens random, and if not, does the directionality of deformation change with geographic and/or stratigraphic position? Do deformed specimens alter the structure of morphospace within groups and/or between groups? Does the inclusion of numerous deformed specimens significantly alter measures of disparity in a study group? Is biological variability overwhelmed by taphonomic variability, and if so, at what taxonomic levels?

Here, we address these questions using a combination of simulation and empirical analyses centered on the abundant dicynodont therapsid Diictodon feliceps. We have four main objectives:Description of the major forms of deformation in the cranium of Diictodon. The range of variation attributable to deformation in Diictodon crania will be described and the possible effects of this deformation on morphometric analysis discussed.

Test the effects of deformation on intraspecific morphospace analyses. An empirical morphospace for Diictodon will be constructed using the results of a principal components analysis of cranial landmark data, and variation in this morphospace as related to biological and deformational variables will be examined. The null hypotheses are that deformation is random and that random deformation adds noise to the data but does not overwhelm known biological signals.

Use simulations featuring known types and amounts of deformation to investigate the circumstances when deformation is likely overwhelming our ability to extract biological signal from a dataset. We will focus on correctly recovering signals reflecting ontogenetic variation and sexual dimorphism (well-supported sources of morphological variability in actual Diictodon fossils; Sullivan & Reisz, 2005). Of particular interest is whether an accurate signal is preserved in simulated datasets displaying levels of variance comparable to the empirical dataset.

Test the effects of deformation on morphospace analyses at higher taxonomic levels. Samples of deformed and undistorted Diictodon specimens will be included in a broad-scale morphospace analysis of Anomodontia, the larger clade to which dicynodonts belong, to determine whether the addition of deformed specimens alters the primary variance structure of the dataset. The overall disparity in the anomodont sample with and without the deformed Diictodon specimens will be compared. The null hypotheses are that deformation is random and that random deformation in one taxon (Diictodon) does not overwhelm interspecific sources of morphological variance.

Materials and Methods

Study system

The Permian dicynodont Diictodon feliceps (Owen, 1876) is a long-ranging species known predominantly from the Karoo Basin of South Africa, although it has also been found in Zambia and China (Angielczyk & Sullivan, 2008; Angielczyk et al., 2014). This species is useful for investigating the effects of deformation in the context of geometric morphometric analysis for several reasons. First, Diictodon is the most abundant terrestrial vertebrate known from the Beaufort Group (Smith, Rubidge & Van der Walt, 2012), making up more than 50% of vertebrate fossils from some localities (Smith, 1993; Sidor & Smith, 2007; Brocklehurst et al., 2017; Day & Rubidge, 2018). Second, the long stratigraphic range of Diictodon (covering five of the eight Beaufort Group assemblage zones, Capitanian to Changhsingian) offers the opportunity to address the potential problem of changing styles of deformation as a result of changes in the sedimentary geology of the Karoo Basin over the course of the Permian (Smith, 1995; Tankard et al., 2009; Barbolini, Bamford & Rubidge, 2016). Third, the sample of South African Diictodon specimens exhibits clear underlying biological variation. Sullivan, Reisz & Smith (2003) demonstrated readily recognizable sexual dimorphism in Diictodon (most notably presence/absence of tusks, but associated with a suite of other cranial characters). Additionally, a nearly complete ontogenetic series is known for this taxon, ranging from probable hatchlings (e.g., SAM-PK-K773 and SAM-PK-K10144) with total skull lengths measuring less than two cm to large adults (e.g., SAM-PK-K6704) with skulls nearly 15 cm in total length. Finally, Diictodon has been the subject of intense taxonomic scrutiny over the past two decades. Several studies based on qualitative comparisons as well as traditional and geometric morphometric analyses have concluded that Diictodon represents a single morphospecies, D. feliceps (King, 1993; Sullivan & Reisz, 2005; Angielczyk & Sullivan, 2008). While additional research exploring the possibility of anagenesis in the lengthy Diictodon record is warranted, taxonomic distinction is unlikely to represent a dominant source of morphological variation in the sample.

Empirical data collection and analyses

A total of 522 crania of Diictodon feliceps (composed of 518 specimens from South Africa, three specimens from Zambia, and one specimen from China) were examined for this study. Of these specimens, 485 were complete enough to have landmarks digitized on at least one side of the skull in dorsal view, with 387 of those complete enough for a bilaterally-symmetric configuration of dorsal landmarks, and 464 specimens were complete enough to have landmarks digitized in lateral view (see associated dataset; Kammerer et al., 2020, https://doi.org/10.5061/dryad.5tb2rbp1x). Specimen images were digitized using ImageJ and TPSDig (Abràmoff, Magalhães & Ram, 2004).

In dorsal view, we digitized a configuration of 16 landmarks consisting of four midline landmarks and six pairs of bilaterally-symmetric lateral landmarks (Fig. 1A): (1) anterior edge of premaxilla; (2 & 11) Prefrontal-lacrimal sutural border at orbital margin; (3 & 12) Anteroventral edge of postorbital bar; (4 & 13) Posteroventral edge of postorbital bar; (5 & 14) Posterior extent of squamosal; (6 & 15) Anteromedial edge of temporal fenestra; (7 & 16) Prefrontal-frontal sutural border at orbital margin; (8) Mid-frontal sutural border with preparietal; (9) Anterior edge of pineal foramen; (10) Mid-parietal sutural border with postparietal.

Figure 1 Landmark configurations utilized in this study.

(A) Dorsal view and (B) lateral view landmarks shown on a largely undistorted skull of Diictodon feliceps (USNM 22949). After reflecting and averaging, the landmarks in the dorsal dataset correspond to landmarks 1–10 in (A). See text for details. Scale bar equals 5 cm. Photos: Christian Kammerer.

We analyzed two permutations of the dorsal view dataset. In the first dataset (“dorsal”), we reflected bilaterally symmetric landmarks across the midline and averaged the positions of the resulting pairs of landmarks to create a series of “half specimens.” In cases where a symmetric landmark was not preserved on one side of the skull, the coordinates of the single preserved landmark were used. This approach follows common practice for dealing with incomplete specimens in paleontological datasets, but the reflecting and averaging process can affect both the biological and taphonomic signals in a dataset including distorted specimens (e.g., elimination of natural asymmetry, or creation of misleading mean values when one side of the skull is highly sheared relative to the other) (Angielczyk & Sheets, 2007). Therefore, we also utilized a second dataset (“dorsal complete”) consisting of only those specimens in which all sixteen landmarks could be digitized. The resulting bilaterally symmetric landmark configurations were then utilized in subsequent statistical analyses without reflecting and averaging symmetric landmarks.

For the lateral view, we digitized 11 landmarks (Fig. 1B): (1) Anteroventral tip of premaxilla; (2) Anterior edge of canine/caniniform process; (3) Septomaxillary-nasal sutural border at narial margin; (4) Posterior edge of canine/caniniform process; (5) Prefrontal-lacrimal sutural border at orbital margin; (6) Prefrontal-frontal sutural border at orbital margin; (7) Ventral edge of maxillary-jugal suture; (8) Postorbital-postfrontal sutural border at orbital margin; (9) Anteromedial edge of temporal fenestra; (10) Posterior extent of squamosal; (11) Posterior edge of parietal. Images of specimens photographed in left lateral view were reflected prior to digitization. If a specimen could be digitized for all lateral landmarks in left and right views, the mean landmark coordinates of the two images were used for subsequent analysis. Otherwise, only the single complete side was utilized.

Specimens were grouped by four variables: sex, size class, assemblage zone (AZ), and deformation style (see associated dataset; Kammerer et al., 2020). Sex was determined by the presence or absence of maxillary tusks, following Sullivan, Reisz & Smith (2003) in considering tusked individuals to be male and tuskless individuals to be female (although accurate identification of whether the tusked cohort represents males or females is not important for the purposes of our analyses, only the existence of a dimorphic pattern in the sample). In the smallest observed tusked specimen of Diictodon (BP/1/102, total skull length 4.87 cm), the tusks are just erupting—all known smaller skulls are tuskless. Because the sex of these presumed juveniles cannot be determined, they were excluded from analyses using this variable. The specimens excluded from analyses of sex make up the “small” size class, consisting of all specimens with a total skull length less than 5 cm. The “medium” size class consisted of specimens with total skull length ranging between 5 and 9 cm, and the “large” size class consisted of specimens with total skull length in excess of 9 cm. Assemblage zone data were available for 419 specimens used in the dorsal analysis, 337 specimens used in the dorsal complete analysis and 391 used in the lateral analysis. Assemblage zone data were derived from specimen labels, Haughton & Brink (1954), Kitching (1977), Smith (1993), and Rubidge (1995). Specimens were considered of “unknown” assemblage zone and excluded from zonal analyses if they lacked locality data altogether, had only vague locality information (e.g., “Cape Province”), or were collected without stratigraphic context at a locality known to span multiple assemblage zones. The largest subset of specimens is from the Tropidostoma AZ, with 216 dorsal, 178 dorsal complete, and 208 lateral landmarked specimens. The second largest sample is from the Cistecephalus AZ, with 112 dorsal, 89 dorsal complete, and 102 lateral. From the Tapinocephalus AZ there are 47 dorsal, 37 dorsal complete, and 40 lateral. From the Daptocephalus AZ there are 24 dorsal, 16 dorsal complete, and 22 lateral. Lastly, from the Pristerognathus AZ there are 20 dorsal, 17 dorsal complete, and 19 lateral specimens. Deformation style was determined qualitatively a priori (see “Patterns of Deformation” below). Many Diictodon skulls are subject to multiple forms of distortion, in which case the dominant style of deformation was given precedence for binning. Undeformed specimens were recognized based on consistent cranial symmetry, uniformity of outline in easily distorted structures like the orbits and foramen magnum, and sutural integrity (i.e., individual cranial bones not pulled apart or subducted beneath others).

Procrustes superimposition and principal components analysis (PCA) of landmark data were performed using the program MorphoJ (Klingenberg, 2008). Meaningful PCA axes were determined using the broken-stick method described by Jackson (1993), which distinguishes between eigenvalues providing significant data structure and those that do not. The digitized specimens of Diictodon listed above were also included in a broad-scale PCA covering all of Anomodontia, with the sample composed of 1876 specimens in dorsal view and 1921 specimens in lateral view (including Diictodon). Landmark protocol for this analysis was identical to that for the within-group Diictodon study, with “dorsal complete” landmarking used for skulls in dorsal view. Anomodont specimens were binned into the higher-level taxa described by Kammerer & Angielczyk (2009) for calculation of within-group means when measuring disparity. Procrustes variance-based morphological disparity was calculated for the anomodont-wide datasets using the “morphol.disparity” function of the R (R Core Team, 2018) package geomorph (Adams & Otárola-Castillo, 2013). The significance of pair-wise differences in disparity among groups was assessed via resampling over 500 iterations.

Procrustes ANOVAs were carried out in geomorph (Adams & Otárola-Castillo, 2013) to assess whether significant amounts of shape variation in our datasets can be ascribed to the four study variables (sex, size class, AZ, and deformation style). The analyses using deformation style as a factor were carried out on the full lateral, dorsal, and dorsal complete Diictodon datasets. In contrast, the analyses focusing on biologically relevant variables (sex, size class, AZ) were carried out using only the specimens in the “undeformed” class from each dataset. When a variable was determined to be significant, pairwise post-hoc tests were carried out to determine whether significant differences in mean shape existed among the classes. Significance was judged at a Bonferroni-corrected alpha level that was appropriate for the number of comparisons.

Simulations

A series of simulations was conducted using the program DefCat (part of IMP; Sheets, 2014), with similar parameters to the simulation studies of Angielczyk & Sheets (2007). DefCat produces simulated deformed and non-deformed datasets in which both deformation and underlying biological signals are known. Our basic protocol consisted of: (1) using non-deformed empirical specimens near the ends of a biological continuum of variation to generate a series of simulated non-deformed specimens that fall along that continuum; (2) creating deformed datasets by using mathematical transformations to apply known types and amounts of deformation to the datasets of simulated non-deformed specimens; and (3) assessing the amount of variance added to the datasets by deformation and testing whether accurate biological signals could be recovered from the deformed datasets.

There were two simulated datasets with known biological signals that were subjected to deformation. The first included a known ontogenetic signal, and the second included a sexual dimorphism signal.

To make the simulated dataset with an ontogenetic signal, two undistorted Diictodon specimens of different sizes were chosen, one a representative “small” individual and one a representative “large” specimen. SAM-PK-K7838 served as the “small” specimen for all of the simulations because it is especially minute (skull length approximately 2.6 cm, near the lower end of known specimens) and because landmarks could be digitized on it in each view. For the dorsal complete dataset, the “large” specimen was SAM-PK-K6041 (skull length approximately 10.1 cm). The “large” specimen in both the dorsal and lateral datasets was USNM 25157 (skull length approximately 11.6 cm; no complete, undistorted specimens near the maximum size for Diictodon, ~15 cm, are known). DefCat was then used to generate as many simulated specimens as we had empirical specimens. This meant 485 dorsal and 464 lateral simulated specimens generated within each simulation. This evenly divides the size and shape differences between the endpoint specimens, creating a simulated ontogenetic series of undeformed specimens. A small amount of identical independent Gaussian noise was added to the data to simulate individual variation among the specimens.

The second series of simulated datasets was created to test the effects of deformation on sexual dimorphism. Here, two datasets were created using sets of either likely male or likely female specimens, so that each dataset would be representative of either male or female Diictodon. Following the hypothesis of Sullivan, Reisz & Smith (2003) that tusked individuals of Diictodon are likely male and tuskless individuals are likely female, two undeformed specimens of either sex were used to create a series of simulated specimens that represent only one sex or the other. For the dorsal complete datasets, BP/1/293 and SAM-PK-K1650 were chosen to represent males, and BSPG 1934-VIII-48 and UCMP 42837 were chosen to represent females. For the dorsal view datasets, NHMUK PV OR 47052 (the holotype of Diictodon feliceps) and SAM-PK-K1650 were chosen to represent males, and BSPG 1934-VIII-48 and TM 299 were chosen to represent females. For the lateral view datasets, NHMUK PV OR 47052 and TM 373 were chosen to represent males, and BSPG 1934-VIII-48 and SAM-PK-K11484 were chosen to represent females. In each case, the number of simulated specimens reflect those in each category for the empirical dataset, resulting in 251 simulated females and 220 males for the dorsal analysis, 194 females and 179 males for the dorsal complete analysis, and 238 females and 214 males for the lateral analysis. A small amount of identical independent Gaussian noise was added to the data to simulate individual variation among the specimens.

To generate our deformed datasets, we used the deformation model described in Angielczyk & Sheets (2007), which applies uniform shear and stretching to a series of landmark configurations. Although shear deformation is only one of several types of deformation observed in specimens of Diictodon (see below), it provides an intuitive starting point for investigating the potential effects of deformation on morphometric data for the species. The deformation model has two main parameters, θ and a. The θ parameter alters the ratio of the long axis to the short axis of the strain ellipse, whereas the a parameter varies the strain ellipse ratio and the orientation of the strain ellipse (details in Angielczyk & Sheets (2007)). In addition to these parameters, the orientation of a specimen’s landmark configuration relative to the strain ellipse will alter the details of its resulting deformation. For example, if the long and short axes of the specimen are aligned with the direction of applied stress, no shearing of the specimen will be apparent, whereas shear will occur when the axes are not aligned. Therefore, an additional parameter of the simulations is the range of angles specimens can take relative to the applied stress, which can be random (i.e., ranging from −180° to 180°) or constrained to a smaller range of angles to produce more stereotyped patterns of deformation in a given dataset.

A total of 18 groups of deformed datasets were generated, nine containing a simulated ontogenetic signal and nine with a simulated sexual dimorphism signal. For a given biological signal, each of the nine groups consisted of 16 individual datasets in which θ = 1.0, 1.5, 2.0, 2.5, 3.0, 3.5, 4.0, 4.5, 5.0, 6.0, 7.0, 8.0, 9.0, 10.0, 15.0, and 20.0, with additional parameters varying from group to group (Table 1). In Group 1, a = 1.0, all specimens were deformed, and the specimens were oriented randomly relative to the direction of applied stress. Groups 2 and 3 had the same parameters as Group 1, except that variable numbers of specimens were left undeformed (50% deformed in Group 2, 94% deformed in Group 3). Groups 4 and 5 had the same parameters as Group 1 except that the orientation of the specimens relative to the strain ellipse was constrained (−45° to 135° in Group 4; 0–90° in Group 5). Groups 6 and 7 had the same parameters as Group 1, but deformation amplitude (a) was allowed to vary (0.95–1.0 in Group 6; 1.0–1.05 in Group 7). All parameters were allowed to vary in Groups 8 and 9 (Table 1).

Table 1 Simulation variables.

	Percentage of deformed specimens	Deformation amplitude	Limited angle variation	Amplitude variation	
Group 1	All	1.0	random	not selected	
Group 2	50% deformed	1.0	random	not selected	
Group 3	94% deformed	1.0	random	not selected	
Group 4	All	1.0	−45° to 135°	not selected	
Group 5	All	1.0	0–90°	not selected	
Group 6	All	0.95–1.0	random	selected	
Group 7	All	1.0–1.05	random	selected	
Group 8	94% deformed	1.0–1.05	−45° to 135°	selected	
Group 9	All	0.95–13.0	0–90°	selected	
Note:

This chart shows the different combinations of variables used for each group in the simulations. For each group, individual datasets were generated for 16 different θ’s (see text for details).

We calculated Procrustes variance-based morphological disparity for all of the simulated deformed datasets using the “morphol.disparity” function of the R package geomorph (Adams & Otárola-Castillo, 2013). These values were then compared to disparity values for the corresponding empirical dataset (e.g., simulated deformed lateral view vs. empirical lateral view) to determine which combinations of deformation parameters produced datasets with levels of disparity comparable to our empirical sample. The significance of differences in disparity was assessed using re-sampling. To determine whether an accurate ontogenetic signal was present in our deformed datasets, we compared the size-correlated shape variation in each deformed dataset to its corresponding undeformed dataset using a homogeneity of slopes test and an ontogenetic trajectory analysis (Adams & Collyer, 2009). An accurate ontogenetic trajectory was considered recoverable if the slope and trajectory parameters for the deformed dataset did not differ significantly from the parameters of the corresponding undeformed dataset. Analyses were also carried out the geomorph package in R (Adams & Otárola-Castillo, 2013).

To test the effects of deformation on the sexual dimorphism signal, we conducted Procrustes ANOVAs in MorphoJ (Klingenberg, 2008) to determine whether simulated male specimens differed significantly in mean shape from simulated females for each set of deformation parameters. Of particular interest in the tests of ontogenetic and sexual dimorphism signals was whether an accurate biological signal could be recovered from the simulated deformed datasets with disparity levels closest to the observed value for the corresponding empirical dataset.

Results

Patterns of deformation

Although the amount of distortion in Diictodon crania exists along a continuum, deformation of Diictodon fossils is not random. The structural properties of the cranium and the limited directionality of compaction of the surrounding sediment limit the range of deformation possible, and nearly all Diictodon skulls can be assigned to one of seven typical styles of deformation (Table 2). Most Diictodon crania are distorted to some degree—undistorted skulls account for only 6% of specimens examined. Two general forms of deformational artifact that can influence morphometric analysis were found in the study sample: artifacts of position and artifacts of perspective. Artifacts of position represent the actual movement of one landmark relative to others as a result of deformation. Artifacts of perspective represent apparent differences in shape between two-dimensional images (e.g., photographs) of a distorted and undistorted specimen in the same orientation. These differences are caused by deformation-induced changes in angulation of various structures in the distorted specimen, resulting in landmarks that appear more distant or closer together in the digitized image than they are in the actual specimen. Each major style of deformation incurs both kinds of artifacts, the details of which are described below.

Table 2 Types of deformation observed in Diictodon feliceps crania.

Type of deformation	Number in sample	Percent of total	
Anteroposterior	22	4.7	
Dorsoventral	149	32.1	
Anterodorsal shear	15	3.2	
Lateral	165	35.6	
Right/left shear	29	6.3	
Saddle-shape	49	10.6	
Undistorted	27	5.8	
Anteroventral shear	8	1.7	
Total lateral specimens	464		
Anteroposterior	23	4.7	
Dorsoventral	161	33.2	
Anterodorsal shear	17	3.5	
Lateral	165	34.0	
Right/left shear	30	6.2	
Saddle-shape	50	10.3	
Undistorted	29	6.0	
Anteroventral shear	10	2.0	
Total dorsal specimens	485		
Anteroposterior	18	4.7	
Dorsoventral	131	33.8	
Anterodorsal shear	15	3.9	
Lateral	120	31.0	
Right/left shear	25	6.46	
Saddle-shape	46	11.9	
Undistorted	25	6.46	
Anteroventral shear	7	1.8	
Total dorsal-complete specimens	387		

The typical styles of deformation in the Diictodon skull are illustrated in Figs. 2–6, and the prevalence of these deformational types is listed in Table 2. The simplest forms of deformation, lateral (Figs. 2C and 2D) and dorsoventral (Figs. 2E and 2F) compression, are the most commonly observed, with each making up roughly a third of the sample. These styles can be related to unidirectional stress, probably the result of compaction of the overlying sediment on skulls buried along their primary planes of rest (i.e., long axis parallel to the substrate, with the skull lying on its lateral, dorsal, or ventral surface). Lateral compression is most strongly evident in dorsal view, and is associated with decreased transverse dimensions of the skull and decreased visible area of the orbits and temporal fenestrae. In lateral view, lateral compression is associated with increased dorsoventral skull height, greater verticality of the postorbital bar (straighter, not as curved as in undistorted skulls), increased visibility of the subtemporal fenestra, and a more perpendicular orientation of the quadrate ramus of the squamosal relative to the long axis of the skull. Conversely, dorsoventral compression is most evident in lateral view, associated with decreased dorsoventral skull height, decreased visibility of the subtemporal fenestra, and posterior angulation of the quadrate ramus of the squamosal. In dorsal view, dorsoventral compression is associated with increased transverse dimensions, increased visible area of the orbits and temporal fenestrae, and greater horizontality of the postorbital (both for the postorbital bar and its contribution to the intertemporal bar). The tusk and caniniform process are the most robust and resistant portions of the Diictodon skull and generally display little change in compressed skulls, but extreme dorsoventral compression can cause these structures to bend anteriorly.

Figure 2 Typical styles of preservation in Diictodon feliceps.

(A) Dorsal and (B) right lateral (mirrored for comparison) views of an undistorted skull (USNM 22949). (C) Dorsal and (D) left lateral views of a laterally compressed skull (USNM 171064). (E) Dorsal and (F) left lateral views of a dorsoventrally compressed skull (SAM-PK-K11558). Scale bars equal 1 cm. Photos: Christian Kammerer.

Figure 3 “Saddle-shape” deformation in therapsid skulls.

(A) Dorsal and (B) right lateral views of a skull of Diictodon feliceps (SAM-PK-K6838). Right lateral views of the theriodont therapsids (C) Ictidosuchoides (Therocephalia; CGS CM86-258) and (D) Cyonosaurus (Gorgonopsia; BP/1/137). Scale bars equal 1 cm. Photos: Christian Kammerer.

Figure 4 Anteroposterior deformation in Diictodon feliceps.

(A) Dorsal and (B) right lateral views of an anteroposteriorly compressed skull (USNM 22948). Scale bar equals 1 cm. Photos: Christian Kammerer.

Figure 5 Sheared skulls in Diictodon feliceps.

(A) Dorsal and (B) right lateral views of an anterodorsally-sheared skull (BP/1/2317). (C) Dorsal and (D) left lateral (mirrored for comparison) views of an anteroventrally-sheared skull (CGS R26). Scale bars equal 1 cm. Photos: Christian Kammerer.

Figure 6 A sheared skull of Diictodon feliceps.

(A) Dorsal, (B) left lateral, and (C) right lateral views of a right/left sheared skull (USNM 25158). Scale bar equals 1 cm. Photos: Christian Kammerer.

More unusual styles of deformation are shown in Figs. 3–6. Of these, the most common (~11% of the sample) is the “saddle-shape” (Fig. 3), in which there is a marked concavity in the dorsal profile of the skull in lateral view. This style of deformation is associated with a seemingly shortened temporal region and elongated snout in dorsal view. However, these apparent differences represent artifacts of perspective in the standard orientation, as the intertemporal bar is angled downwards (making it appear shorter) and the snout is angled upwards (making it appear longer, because the anterior edge of the premaxilla, normally directed downwards and thus not visible in dorsal view, is exposed dorsally). The “saddle-shape” can best be interpreted as a variant style of dorsoventral compression related to the inherent architectural properties of the dicynodont cranium. Although the dicynodont snout and posterior braincase are robust, rigid structures (especially with the massive canine roots associated with the former in most species), the midsection of the skull features relatively weak supporting elements (the postorbital bars and the comparatively delicate midline ossifications of the braincase) between two major areas of soft tissue (the temporal musculature and eyes). Thus, in the presence of compressive stress in the dorsoventral plane, it is expected that the midsection of the skull would yield before the anterior or posterior regions, creating the “saddle-shape”. Supporting this interpretation, “saddle-shaped” skull distortion is also observed (Figs. 3C and 3D) in all theriodont therapsid groups (in which the snout/canine region is also generally more robust than the orbitotemporal region), but not tapinocephalid dinocephalians, which possess massively pachyostosed postorbital bars, relatively small temporal fenestrae and orbits, and internal skull construction optimized for resisting forces applied to the dorsal surface of the skull (Barghusen, 1975; Benoit et al., 2017).

USNM 22948 (Fig. 4) represents a rare (~5% of the sample) case of anteroposterior compression. In this specimen, failure under compression occurred around the pineal foramen, such that the parietals were subducted under the frontals. Some cases of anteroposterior compression (e.g., BP/1/292) also exhibit the “saddle-shape”, probably as a result of the skull bending in on itself under stress. The rarity of anteroposteriorly compressed specimens in the sample can probably be attributed to the unusual circumstances required for its occurrence, necessitating either burial of a skull not on its primary plane of rest (i.e., long-axis perpendicular to the substrate) or tectonic influences (e.g., folding).

In addition to compression, various forms of deformation attributable to shear are observed in the study sample. Most distorted dicynodont specimens exhibit some degree of asymmetry as a result of shear (note the relative positions of the orbits and squamosals in Fig. 2C and postorbital bars in Fig. 4A), although complexly deformed skulls exhibiting marked distortion along multiple shear axes are rare. Of the varieties of shear-related cranial deformation, the most problematic for morphometric purposes is anterodorsal or “face forward” shear (~3.5% of the sample), as illustrated by BP/1/2317 (Figs. 5A and 5B). In this specimen, the dorsal surface of the skull has been sheared forward, resulting in an artifact of perspective (shortened snout) in dorsal view and actual positional changes (circumorbital landmarks shifted anterior to the maxillary landmarks) in lateral view. Anteroventral (~2% of the sample; Figs. 5C and 5D) and right/left (~6% of the sample; Fig. 6) sheared skulls are also observed in the sample.

In terms of the effects of deformation on the cranial landmarks used in this analysis, each major style involves characteristic alterations of different suites of landmarks (although degree of change varies with each specimen). For lateral compression, the primary positional artifacts in dorsal view are medial movement of the set of landmarks along the lateral skull surface (2/11, 3/12, 4/13, and 5/14), bringing them physically closer to the midline landmarks 8, 9, and 10. Greater proximity of landmarks along the ventral margins of the orbit and temporal fenestra (2/11, 3/12, and 4/13) to those on the dorsal margins (6/15, 7/16) is also observed, but this can be attributed largely to perspective artifacts related to verticalization of the postorbital bar and orbit. For example, in lateral view, landmarks 4/13 and 6/15 of a laterally compressed skull are separated by either the same distance as in undistorted skulls or a slightly greater distance (related to general increase in skull height), but in dorsal view the straightening of the postorbital bar puts these landmarks in nearly the same plane (as the natural lateral curvature of the postorbital bar is lost), sharply decreasing the distance between them in the projected image.

For dorsoventral compression, most of the changes in landmark position are the opposite of those for lateral compression (e.g., in dorsal view, the skull edge landmarks move laterally relative to the midline with increasing transverse dimensions of the skull), but with some peculiarities. In lateral view, greater proximity between the set of landmarks along the base of the skull (1, 7, and 10) and those higher up (3, 5, 6, 8, 9, and 11) in projected images represents a combination of artifacts of perspective (e.g., greater horizontality of the postorbital bar making landmark 7 appear closer to 8 and 9) and position (the more dorsal series of landmarks has physically moved ventrally as a result of compression). Anterodorsal displacement of landmarks 2 and 4 is also sometimes observed, in the extreme cases where the tusk or caniniform process has bent anteriorly.

As noted above, “saddle-shape” deformation results in decreased intertemporal length and increased snout length in dorsal view (artifacts of perspective), bringing the midline landmarks either closer together (9 relative to 10) or further apart (8 relative to 1) than in an undistorted skull. In lateral view, the most affected landmarks are those around the dorsal margin of the orbit (6, 8, and 9), which become displaced ventrally as a result of the collapse of the skull roof and postorbital bar. For anteroposterior compression, the primary change in both dorsal and lateral views is reduction in distance between all landmarks along the long axis of the skull. In lateral view, the skull also appears taller than in undistorted specimens (similar to the condition in laterally compressed specimens), resulting in greater separation between ventral (1, 2, 4, 7, and 10) and dorsal (3, 5, 6, 8, 9, and 11) landmarks.

Anterodorsal shear results in a shorter observed snout in dorsal view (bringing landmark 8 closer to 1, the opposite of the “saddle-shape” condition), but no change in observed length of the intertemporal bar (as the entire structure is displaced forwards, so the relative positions of landmarks along its length remain unchanged). In lateral view, dorsal landmarks (3, 5, 6, 8, 9, and 11) are generally displaced anteriorly relative to the ventral (1, 2, 4, 7, and 10) ones. The opposite is the case for anteroventral shear. Left/right shear results in variable landmark movement in lateral view, depending on which side of the skull is being digitized. Dorsally, it results in displacement of the pairs of symmetric landmarks on opposing sides of the skull relative to one another, a result only captured in the dorsal complete analysis.

Principal components analysis of Diictodon feliceps crania: Lateral

Although they account for just 49.8% of the cumulative variance in the data set, only the first two lateral PC axes contain meaningful structure under the broken-stick criterion. PC1 accounts for 26.6% of variance in the lateral dataset. This PC describes relative skull height, snout depth, and angulation of the temporal region (Fig. 7). Specimens with positive scores on PC1 exhibit tall skulls relative to length, deep snouts, and short temporal regions with less oblique angles between the temporal landmarks (9 and 11). PC2 accounts for 23.2% of variance in lateral data. This PC primarily describes angulation of the face and temporal fenestra. In specimens with positive scores on PC2, the face is angled forwards, such that circumorbital landmarks (5, 6, 8) are more anterior and maxillary landmarks (2, 4) are more posterior than in the mean configuration.

Figure 7 Morphological variation described by the two principal component (PC) axes from the empirical analysis of lateral Diictodon data.

The light blue wireframe represents the mean landmark configuration and the dark blue wireframe represents the changes in landmark position associated with a positive score on that PC. (A) PC 1. (B) PC2.

Deformation style is the only variable that shows clear structure in the PC1 vs. PC2 morphospace (Fig. 8B). Undistorted skulls cluster around the origin. Laterally and anteroposteriorly compressed specimens generally have high positive scores on PC1, whereas dorsoventrally compressed and “saddle-shaped” specimens generally have high negative scores on this PC, although there is significant overlap between these clusters at the origin. Anteroventrally and right/left sheared specimens also tend to occupy the positive region of PC1, although there are several marked outliers for the latter. The majority of anterodorsally sheared specimens occupy a central position on PC1, with a few outliers. Direction of shear is the dominant source of structure for PC2. Anteroventrally sheared specimens range from a central position to highly negative and all anterodorsally sheared specimens are positive on PC2. Representatives of the other major styles of deformation vary widely on PC2, although the majority of “saddle-shaped” specimens occupy the negative region, a result attributable to the anterodorsal angulation of the caniniform process relative to the orbits when a skull is bent at the midsection. The other variables (size class, sex, and assemblage zone) exhibit random occupation of PC1 vs. PC2 morphospace (Fig. 8).

Figure 8 PC1 vs. PC2 morphospace plots of lateral empirical data.

Plots show variation in (A) sex, (B) deformation style, (C) size class, and (D) assemblage zone. Specimens for which sex or assemblage zone is unknown were included in the analysis but are uncolored in the plots.

Principal components analysis of Diictodon feliceps crania: Dorsal

Only the first two PC axes derived from the dorsal data set contain significant structure under the broken-stick criterion, and together they account for 58.3% of cumulative variance in the data. PC1 accounts for 38.4% of variance in the dorsal data. This PC describes snout length and position of the postorbital bar (Fig. 9). Specimens with positive scores on PC1 exhibit relatively short snouts and an anterior position for the dorsal limit of the postorbital bar (Landmark 6) relative to the mean configuration. PC2 accounts for 19.9% of variance in the dorsal data. This PC primarily describes skull breadth. Specimens with positive scores on PC2 exhibit transversely narrow skulls in which the posterior tip of the intertemporal bar (Landmark 10) is situated posterolateral to the mean configuration.

Figure 9 Morphological variation described by the two principal component (PC) axes from the empirical analysis of dorsal Diictodon data.

The light blue wireframe represents the mean landmark configuration and the dark blue wireframe represents the changes in landmark position associated with a positive score on that PC. (A) PC1. (B) PC2.

As with the lateral data, the only variable showing clear structure in PC1 vs. PC2 morphospace is deformation style (Fig. 10B). Direction of shear is associated with PC1, with anterodorsally sheared specimens occupying positive PC1 space and anteroventrally sheared specimens occupying neutral to negative PC2 space (with one outlier). Most other styles of deformation show no pattern of association with PC1, although the majority of anteroposteriorly compressed specimens also have a positive score on this PC. Direction of compression is associated with PC2. Laterally compressed specimens are generally positive on PC2, whereas dorsoventrally compressed specimens are mostly negative. Anterodorsally sheared specimens also show a notable cluster in negative space on PC2. The other variables (size class, sex, and assemblage zone) exhibit random occupation of PC1 vs. PC2 morphospace (Fig. 10).

Figure 10 PC1 vs. PC2 morphospace plots of dorsal empirical data.

Plots show variation in (A) sex, (B) deformation style, (C) size class, and (D) assemblage zone. Specimens for which sex or assemblage zone is unknown were included in the analysis but are uncolored in the plots.

Principal components analysis of Diictodon feliceps crania: Dorsal Complete

Only the first two PC axes derived from the dorsal complete data set contain significant structure under the broken-stick criterion, and together account for 47.0% of the variance in the data. PC1 accounts for 29.1% of the variance in the dorsal complete data. PC1 and PC2 both describe information about length and breadth of the skull, though in different ways. PC1 describes differences in anteroposterior length, showing changes associated with shortening of the back of the skull for specimens with high positive scores (Fig. 11A). It also shows changes in the transverse breadth of the skull, with transversely wider specimens scoring higher on this axis. PC2, on the other hand, captures information about length change near the front of the skull. Specimens with high positive scores in PC2 have relatively shortened snouts compared to the mean configuration (Fig. 11B). Specimens scoring high on PC2 are also transversely wider than the mean shape of the dataset in the temporal region, but narrower in the snout.

Figure 11 Morphological variation described by the two principal component (PC) axes from the analysis of dorsal complete Diictodon data.

The light blue wireframe represents the mean landmark configuration and the dark blue wireframe represents the changes in landmark position associated with a positive score on that PC. (A) PC1. (B) PC2.

Like the lateral and dorsal data, the predominant variable showing clear structure in PC1 vs. PC2 morphospace for dorsal complete is deformation style (Fig. 12B). Laterally compressed specimens tend to have low to moderate scores on PCs 1 and 2, whereas dorsoventrally compressed specimens tend to have higher scores on both PC axes. Anteroposteriorly deformed specimens generally have high positive scores on PC1 and anterodorsally sheared specimens generally have high positive scores on PC2. Assemblage zone data, skull size and sex show largely random distribution in the morphospace (Fig. 12). Specimens from the Daptocephalus AZ do tend to mostly have higher values on PC1 and lower values on PC2, but they are broadly overlapped by specimens from the other assemblage zones on these axes.

Figure 12 PC1 vs. PC2 morphospace plots of dorsal complete empirical data.

Plots show variation in (A) sex, (B) deformation style, (C) size class, and (D) assemblage zone. Specimens for which sex or assemblage zone is unknown were included in the analysis but are uncolored in the plots.

Procrustes ANOVAs

Deformation style was found to explain a significant amount of variation in all three empirical datasets, but significant results were rarer for the biologically relevant factors (see associated dataset; Kammerer et al., 2020, https://doi.org/10.5061/dryad.5tb2rbp1x). Only size class was found to be significant in the dorsal (p = 0.021) and dorsal complete (p = 0.048) datasets (undeformed specimens only), although sex and AZ approached significance in some cases (p-values range between 0.071 and 0.622). The pairwise comparisons revealed that the different deformation styles usually have significantly different mean shapes (20 out of 21 comparisons significant for the dorsal, dorsal complete, and lateral data), and also differ significantly from the undeformed specimens in the majority of cases (4 out of 7 comparisons significant for the dorsal and dorsal complete data, 5 out of 7 for the lateral), even under the rather stringent Bonferroni-corrected alpha level of 0.002. The cases where the mean shapes did not significantly differ are generally logical when one considers the way in which the deformations styles tend to alter specific aspects of shape. For example, laterally compressed and right/left sheared specimens do not have mean shapes that differ significantly from undeformed specimens in the lateral view datasets because these deformation types tend to have relatively mild effects on shape when viewed from the side. Among the pairwise comparisons for the size classes, only medium and small specimens in the dorsal complete dataset had a significantly different shape at the Bonferroni-corrected alpha of 0.02.

Analysis of Diictodon within Anomodontia

For the all-anomodont lateral PCA including only undistorted Diictodon specimens, the first four PC axes account for 70.6% of cumulative variance in the data. PC1 accounts for 29.2% of variance and describes general skull height, snout depth, and temporal fenestra length (Fig. 13A). Specimens with positive scores on PC1 have relatively low skulls, shallow snouts, and long temporal fenestrae. The majority of anomodont diversity occupies positive PC1-space, with high negative scores largely restricted to the extremely deep-snouted lystrosaurids (Fig. 14). PC2 accounts for 17.7% of variance in the data and describes relative snout length, orbital height, and temporal/occipital angulation (Fig. 13B). Emydopoids (especially the fossorial cistecephalids) generally have high positive scores on PC2 and are characterized by very short snouts, large orbits, and a tall occiput with the posterior edge of the parietal located slightly anterodorsal to the posterior edge of the squamosal. Kannemeyeriiforms, rhachiocephalids, and some basal dicynodontoids generally have high negative scores on PC2 and are characterized by long snouts, small orbits relative to skull size, and a very long temporal region where the posterior edge of the parietal is located far anterior to the posterior edge of the squamosal. PC3 accounts for 13.6% of variance in the data and describes angulation of the postorbital bar and height of the temporal fenestra (Fig. 13C). Cistecephalid emydopoids have high negative scores on PC3 and are characterized by an anterior position for the junction between the postorbital bar and zygoma and tall temporal fenestra. Finally, PC4 accounts for 10.0% of variance in the data and describes orbital width and maxillary/tooth row length (Fig. 13D). Non-dicynodont anomodonts (“dromasaurs” and venyukovioids) have high positive scores on PC4 and are characterized by very wide orbits and long snouts with marginal dentition (as opposed to the single tusk and/or few postcanines of most dicynodonts). Undistorted Diictodon specimens occupy a location near the origin in all significant PC plots (Fig. 14).

Figure 13 Morphological variation described by the four principal component (PC) axes from the analysis of lateral anomodont data.

The light blue wireframe represents the mean landmark configuration and the dark blue wireframe represents the changes in landmark position associated with a positive score on that PC. (A) PC1. (B) PC2. (C) PC3. (D) PC4.

Figure 14 Anomodont lateral morphospace including only undistorted Diictodon.

The results of principal components analyses, showing the four primary axes of variation. (A) PC1 vs. PC2. (B) PC1 vs. PC3. (C) PC1 vs. PC4.

The results of the all-anomodont lateral PCA including deformed Diictodon specimens were similar to the previous analysis. The first four PCs account for a cumulative 70.1% of variance in the data, with PC1 contributing 28.5%, PC2 contributing 17.7%, PC3 contributing 13.7%, and PC4 contributing 10.3%. The morphological differences described by these PCs are the same as in the previous analysis, and morphospaces constructed using these PC axes are generally similar to those illustrated above (Fig. 15). Deformed Diictodon specimens occupy a much broader range of morphospace than in the previous analysis, however, and the Diictodon cluster overlaps much of anomodont diversity on PCs 2 and 3, with a few outliers even impinging on lystrosaurid negative space on PC1 (Fig. 16). Procrustes variance-based morphological disparity for Anomodontia (calculated as was done for the simulated deformed datasets, using the “morphol.disparity” function of the R package geomorph (Adams & Otárola-Castillo, 2013)) does not appreciably differ between the data containing only undistorted (0.0334) and all Diictodon specimens (0.0336), however.

Figure 15 Anomodont lateral morphospace including deformed Diictodon.

The results of principal components analyses, showing the four primary axes of variation. (A) PC1 vs. PC2. (B) PC1 vs. PC3. (C) PC1 vs. PC4.

Figure 16 Position of Diictodon specimens (in blue) in anomodont lateral morphospace based on undistorted (left column) and all (right column) specimens.

(A) PC1 vs. PC2 including only undistorted Diictodon specimens. (B) PC1 vs. PC2 including deformed Diictodon specimens. (C) PC1 vs. PC3 including only undistorted Diictodon specimens. (D) PC1 vs. PC3 including deformed Diictodon specimens. (E) PC1 vs. PC4 including only undistorted Diictodon specimens. (F) PC1 vs. PC4 including deformed Diictodon specimens.

For the all-anomodont dorsal complete PCA including only undistorted D. feliceps specimens, the first four PC axes account for 71.1% of cumulative variance in the dataset. The first three PC axes show strong phylogenetic signal based on qualitative observation of the morphospace. PC 1 accounts for 39.0% of variance in the data and describes relative skull length (Fig. 17). The extremely short-skulled cistecephalid emydopoids and lystrosaurids have high positive scores on PC1 (Fig. 18). PC2 accounts for 14.7% of variance in the data and describes angulation of the orbits and temporal region. Cistecephalid emydopoids have high positive scores on PC2 and are characterized by forward-facing orbits and squared-off temporal regions in which the posterior edges of the squamosals occupy the same transverse axis of the skull as the posterior edge of the parietal. PC3 accounts for 11.6% of variance in the data and describes temporal fenestra length and breadth. Cryptodonts generally have high positive scores on PC3 and are characterized by very broad temporal fenestrae, whereas most emydopoids have negative loadings on PC3 and relatively narrow fenestrae in dorsal view. Diictodon occupies low negative space on PC1 and 2 and is broadly distributed on PC3. PC4 accounts for 5.6% of variance in the data and describes the effects of right/left shear (Fig. 17). Positive scores on PC4 are associated with a relatively posterior position for landmarks on the left side of the skull and anterior position for those on the right; negative scores are the opposite. This result indicates that even though phylogenetic signal accounts for the majority of variance in higher-level analyses of Anomodontia, and even with the large sample of deformed Diictodon specimens excluded, there is still a significant taphonomic signal in this data. An unusually high number of kannemeyeriiforms occupy positive PC4 space, and a number of cryptodonts and basal dicynodontoids also occupy this space (Fig. 18). These three groups represent the anomodonts with the proportionally largest temporal fenestrae. Although all lateral skull margin landmarks show movement from the mean on PC4, the greatest magnitude of change is along the landmarks surrounding the temporal fenestra (4/13, 5/14). Given that the subtemporal bar bounding the temporal fenestra is a relatively thin structure in these taxa, they are probably disproportionately susceptible to high degrees of deformation in this region.

Figure 17 Morphological variation described by the four principal component (PC) axes from the analysis of dorsal complete anomodont data.

The light blue wireframe represents the mean landmark configuration and the dark blue wireframe represents the changes in landmark position associated with a positive score on that PC. (A) PC1. (B) PC2. (C) PC3. (D) PC4.

Figure 18 Anomodont dorsal complete morphospace including only undistorted Diictodon.

The results of principal components analyses, showing the four primary axes of variation. (A) PC1 vs. PC2. (B) PC1 vs. PC3. (C) PC1 vs. PC4.

Including deformed Diictodon specimens in the all-anomodont dorsal complete analysis yields similar primary sources of variation as in the prior analysis. The first four PCs account for a cumulative 70.1% of variance in the data, with PC1 contributing 37.9%, PC2 contributing 15.3%, PC3 contributing 10.3%, and PC4 contributing 6.6%. The addition of deformed Diictodon specimens alters the shape of anomodont morphospace, but does not appreciably change its structure (i.e., the same taxa occupy the same regions of morphospace; Fig. 19). As for the lateral analysis, the deformed specimens occupy a much broader range of morphospace than observed for this taxon when only undistorted specimens are included (Fig. 20). This is also reflected in the Procrustes variance for undistorted vs. all Diictodon specimens (0.222 vs. 0.428 respectively). Procrustes variance for Anomodontia as a whole does not change significantly whether only undistorted or all Diictodon specimens are included, however (0.2741 vs. 0.2702 respectively).

Figure 19 Anomodont dorsal complete morphospace including deformed Diictodon.

The results of principal components analyses, showing the four primary axes of variation. (A) PC1 vs. PC2. (B) PC1 vs. PC3. (C) PC1 vs. PC4.

Figure 20 Position of Diictodon specimens (in blue) in anomodont dorsal complete morphospace based on undistorted (left column) and all (right column) specimens.

(A) PC1 vs. PC2 including only undistorted Diictodon specimens. (B) PC1 vs. PC2 including deformed Diictodon specimens. (C) PC1 vs. PC3 including only undistorted Diictodon specimens. (D) PC1 vs. PC3 including deformed Diictodon specimens. (E) PC1 vs. PC4 including only undistorted Diictodon specimens. (F) PC1 vs. PC4 including deformed Diictodon specimens.

Simulations

We calculated the morphological disparity present in each of our simulated deformed datasets and mapped it to observed disparity in the empirical datasets. For the first four sets of parameters in the dorsal and lateral datasets, and the first five in the dorsal complete dataset, disparity in the simulated datasets was lower than in the empirical datasets at low values of θ and higher at high θ values. However, simulated disparity was close to empirical at values of θ in the 7.0–15.0 range, with the greatest similarities usually in the range of 8.0–10.0. These simulated datasets did not show significantly different disparity from the observed empirical datasets. For parameter sets 6–9 simulated disparity was always higher than in the empirical datasets. We consider the simulated datasets with disparities close to empirical levels to be the most relevant for further tests of whether we can accurately recover biological signals from datasets including deformed specimens, with simulated datasets with higher or lower disparity levels representing “worst case” and “best case” scenarios, respectively.

In the trajectory analysis, each simulated deformed dataset was compared to its simulated non-deformed counterpart. Specifically, we considered three parameters of the ontogenetic trajectories: (1) their lengths (trajectory size); (2) their orientations in relation to each other (trajectory orientation; measured by principal vector correlation); and (3) the paths their trajectories take through shape space (trajectory shape) (Adams & Collyer, 2009).

For the majority of the datasets, there was no significant difference between the ontogenetic trajectories of the deformed and non-deformed datasets. The aspect that showed the largest difference between trajectories was trajectory size. In the lateral datasets, 6.7% of the datasets showed significant differences in disparity between empirical and simulated datasets. In the dorsal datasets, 9% showed significant differences, and in the dorsal complete datasets, 5.8% showed significant differences. Combined, 21.5% of the datasets showed significant differences in trajectory size between the empirical and simulated datasets, and 78.5% showed no significant difference. In the lateral datasets, differences were most often found between θ 1.0 and 2.5 and above θ 6.0, but was less common between θ 2.5 and 6.0. Differences were found for each parameter group except for parameter group 2, and were most often seen in groups 6–9. In the dorsal and dorsal complete datasets, differences were found for each parameter group except for group 5. Just as with lateral, differences were most often found in parameter groups 6–9.

The second most common difference was trajectory shape. In the lateral datasets, 4.6% of the data showed significant differences between the non-deformed and deformed trajectory shapes. In dorsal, 3.5% showed significant differences, and in dorsal complete, 4.4% showed significant differences. In total, 12.5% of the data showed significant differences in trajectory shape, with 87.5% showing insignificant differences. In the lateral datasets, differences were found for parameter groups 3 and 5–9, but not 1, 2, or 4. For dorsal, differences were found in parameter groups 4 and 6–9, but not 1–3 or 5. In dorsal complete, differences were found in all parameter groups except group 2. As with trajectory size, trajectory shape showed most prevalent difference in parameter groups 6–9.

Significant differences in trajectory orientation were very uncommon. There were only three instances of significant differences in trajectory orientation in the lateral view datasets (θ 1.0 in parameter sets 7 and 9, and θ 10.0 in parameter set 8), two instances in the dorsal view datasets (θ 3.5 in parameter set 8 and θ 10.0 in parameter set 9), and none in the dorsal complete datasets.

The analytical procedure for the homogeneity of slopes test was the same as for the ontogenetic trajectory analysis, with the slope of the deformed dataset being compared to that of its non-deformed counterpart. The results of the analysis were similar to those of the trajectory analysis, with significant differences in slope rarely occurring between simulated deformed and non-deformed datasets. In the lateral datasets, significant differences occurred only 3 times. In dorsal, significant differences occurred 10 times, and in dorsal complete, significant differences occurred 11 times. In total, significant differences were found in 5.5% of all of the datasets, and there was no significant difference in slope for 94.5% of the datasets. There was no real trend for the thetas at which these differences would occur, but it was more common in parameter groups 5–9 in the dorsal and dorsal complete datasets. Taken together, these results imply that it may be possible to recover an accurate ontogenetic signal from therapsid datasets that include deformed specimens, even in cases where the amount of disparity contributed by deformation exceeds what is observed in the empirical Diictodon dataset.

Our analysis of the effects of deformation on a sexual dimorphism signal focused on whether we could recover a significant difference in shape between sets of simulated deformed male and female specimens. The results of our Procrustes ANOVAs comparing deformed male and female datasets were very straightforward: a significant difference between males and females was preserved in all of the deformed datasets, regardless of the starting deformation parameters. This finding indicates that it may be possible to identify instances of sexual dimorphism in therapsid morphometric datasets even when deformed specimens are included.

Discussion

The results of the principal components analyses of Diictodon crania strongly indicate that deformation is the dominant source of morphological variance in the lateral, dorsal, and dorsal complete empirical datasets. The Procrustes ANOVA results corroborate these observations, with deformation always having a significant effect, and deformation styles having significantly different mean shapes in the majority of cases. Deformation style exhibits considerable structure in PC1 vs. PC2 morphospace, with almost all major styles of deformation occupying a characteristic region of morphospace. Left/right shear is the sole exception to this pattern, with specimens more evenly distributed in morphospace. This likely stems in part from averaging of symmetric landmarks during the construction of the lateral and dorsal datasets, which is expected to remove some effects of shear deformation (Angielczyk & Sheets, 2007). However, the pattern is also apparent in the dorsal complete dataset, which does not have averaged landmark coordinates. We hypothesize that this is because specimens can be affected by shearing in different ways (e.g., “left side forward” vs. “right side forward”), resulting in skull shapes that do not cluster together in morphospace. The structuring of specimens in morphospace according to their deformation style is similar to that observed in an analysis of deformed skulls of the dinosaur Psittacosaurus (Hedrick & Dodson, 2013), suggesting that this pattern may be ubiquitous in fossil skull datasets with relatively large sample sizes. Hedrick et al. (2019) recently demonstrated substantial morphospace dispersion of deformed Psittacosaurus postcrania as well, regardless of base morphology (i.e., flat vs. columnar) of the bones in question. Although architectural properties of elements clearly have some influence on proclivity to distortion and the effects thereof (see “Patterns of deformation” above and discussion by Tschopp, Russo & Dzemski (2013)), deformation appears to represent an important factor structuring morphospace across elements in fossil specimens.

In contrast to the strong signal from deformation style, Diictodon specimens categorized by the biological variables of sex and size class (a proxy for age) exhibit seemingly random occupation of morphospace. Krone, Kammerer & Angielczyk (2019) were able to recover an ontogenetic signal for Diictodon skulls in lateral view using a smaller sample of more or less undeformed specimens and a slightly different configuration of landmarks in lateral view. The absence of an obvious size-shape relationship in the empirical data here suggests that it was overprinted by deformation, mirroring the conclusion of Hedrick & Dodson (2013) that their Psittacosaurus dataset did not preserve an original allometric signal. It also appears that size class and sex have only a weak effect on overall skull shape in Diictodon under even the best circumstances. These factors typically explained only about 10–15% of the shape variance among the undeformed specimens in our datasets.

Although mostly random, some weak structure is observed when specimens are categorized by assemblage zone in the PC plots, and assemblage zone also explained about 20% of the shape variance among our undeformed specimens in the Procrustes ANOVAs. Most Daptocephalus AZ specimens displayed lower scores on PC2 for the dorsal and dorsal complete datasets, and mostly higher scores on PC1 for the dorsal complete dataset. These positions can be accounted for by the absence of any anterodorsally sheared specimens in the Daptocephalus AZ sample. We interpret the lack of anterodorsal deformation among Daptocephalus AZ specimens to be an artifact of small sample size, because anterodorsal shear is one of the rarest styles of deformation (albeit one with an outsized influence on morphospace), and the Daptocephalus AZ is the second smallest sample. Moreover, several specimens of the Daptocephalus AZ index taxon Daptocephalus leoniceps clearly exhibit anterodorsal shear (e.g., BP/1/555, NMQR 960), allowing us to reject a change in basin-wide deformation style as the source of the pattern. The morphometric heterogeneity of specimens from the different assemblage zones provides some corroboration of the hypothesis that only a single species of Diictodon (D. feliceps) is present in the Karoo Basin (Sullivan, Reisz & Smith, 2003; Sullivan & Reisz, 2005; Angielczyk & Sullivan, 2008), but this conclusion is tempered by the possibility that deformation might have overprinted an underlying taxonomic signal in the data. Furthermore, the presence of low-level variance tied to assemblage zone even among undeformed skulls suggests there could be merit in exploring possible anagenetic variation in the Diictodon record.

Anomodont morphospace is dominated by phylogenetic signal in the sense that specimens belonging to the higher-level clades used in binning tend to cluster together. However, the relationships between these clades (as recovered in recent phylogenetic analyses, for example, Angielczyk & Kammerer, 2017; Kammerer et al., 2019) are not clearly reflected by their relative positions in morphospace. Undistorted Diictodon specimens occupy positions near the mean on PCs 1–3 in the lateral analyses, and deformed specimens are dispersed widely around this region in all directions. In the dorsal analysis, undeformed specimens tend to have negative scores on PCs 1–3, and again the addition of deformed specimens causes the occupied area to expand uniformly in all directions from this starting point. In each case, essentially the same PC axes are recovered regardless of deformation in Diictodon, indicating that for the analysis of total occupied morphospace at broad taxonomic scales, inclusion of deformed specimens is not misleading. Within anomodont morphospace, however, it is clear that the extent occupied by Pylaecephalidae (and Diictodon feliceps specifically) has been greatly expanded by the deformed specimens, with especially troubling implications for between-group measures of disparity at lower taxonomic levels.

Given the strong apparent phylogenetic signal in anomodont morphospace, the inclusion of deformed taxonomic singletons is warranted in clade-level analyses of morphological disparity. Because undistorted specimens are so rare (<6% of the sample in the case of heavily-sampled Diictodon), the likelihood of anomodont singletons preserving the undistorted cranial morphology of their species is low. Nevertheless, our morphospace results indicate that even if a deformed specimen does not occur precisely where it would if it was not deformed, the majority of deformed specimens still fall in the region occupied by their larger clade. This result is encouraging for analyses that seek to maximize taxonomic inclusivity. It is notable, however, that even at this broad scale, some variation due to taphonomic overprinting can be discerned, with PC4 from the all-anomodont dorsal complete analysis being related to specimen shear.

Like the empirical datasets, deformation increased the disparity of the simulated datasets. However, in most or all cases we were able to recover an accurate (simulated) ontogenetic or sexual dimorphism signal from the simulated datasets, whereas there was no obvious structure associated with these factors in the empirical datasets that included deformed specimens. A possible explanation for this difference in the empirical and simulation results is the fact that the simulations utilized only a single kind of deformation (shear), in contrast to the multiple types of deformation experienced by the empirical specimens. The latter case might overprint the shapes of specimens to a greater degree than a more stereotyped deformation style, even if the various deformation styles tend to segregate into specific regions of morphospace. Therefore, although the simulation results raise the possibility that it might be possible to recover accurate biological signals from datasets including deformed specimens, this likely will be difficult in practice. One potential approach for trying to extract biological information from deformed specimens would be to conduct the analyses on groups of specimens sharing a similar deformation style, which might result in datasets more similar to our simulated data. Somewhat counterintuitively, it seems this is a case where a more restricted sample is more likely to recover biological signal. However, even a restricted sample could be misleading if the direction of deformation overlaps with the biological signal. As an example, Jasinoski & Abdala (2017) recognized sexual dimorphism in the Triassic cynodont Galesaurus, with one morph characterized by a generally broader skull than the other (especially evident in the transverse dimensions of the snout and zygomatic arches). This pattern was observed in a number of undistorted skulls in association with other features, suggesting that it is not the result of a taphonomic overprint. In a sample of only dorsoventrally or laterally compressed skulls, however, this pattern would not be discernible with confidence, given the strong influence of these deformation styles on skull breadth.

Alternatively, given that deformation in both our simulated and empirical datasets seems to add variance in a roughly even fashion in all directions in morphospace, it may be that our simulated datasets have clearer biological signals than would be the case in the empirical data even if deformation was absent. We used empirical specimens as the starting points for our simulated datasets, but the specimens chosen were relatively extreme shapes (e.g., very large and very small undistorted empirical specimens for the simulated datasets with an ontogenetic signal), which could result in an exaggerated “biological” signal. However, the simulated ontogenetic signal in our lateral view dataset is quite similar to that documented by Krone, Kammerer & Angielczyk (2019) for Diictodon (larger specimens have proportionally deeper snouts and are more dorsoventrally constricted near the level of the orbits), so we do not think we have seriously mischaracterized the biological signals included in the simulated datasets. Instead, we consider it more likely that better preserved biological signals in the simulated deformed datasets is a reflection of the simpler style of deformation applied in the simulations.

A final aspect of both our empirical and simulated datasets that warrants discussion is their comparatively large sample size. Diictodon is the single most common dicynodont in the Permian rocks of the Beaufort Group in the Karoo Basin (Smith, Rubidge & Van der Walt, 2012), and the very high percentage of deformed specimens in the empirical dataset underscores how ubiquitous taphonomic deformation is in Karoo fossils. The large sample also facilitates a detailed characterization of different deformation styles, and helps to show that biological signals such as ontogeny and sexual dimorphism are not major sources of variation in the dataset compared to the effects of deformation. However, most Karoo synapsid taxa are represented by fewer (often many fewer) specimens. Given that Diictodon shows several stereotyped styles of deformation that differ considerably in shape, it is easy to see why the effects of deformation have been such a confounding factor in Karoo synapsid taxonomy, particularly in cases where species are represented by only a handful of specimens. Recent taxonomic work has been more circumspect about the potential effects of deformation, and our finding that deformation can overprint biological signals indicates that similar care is necessary in studies that seek to quantify morphological variation. More optimistically, it is worth noting that the Diictodon specimens studied here are quite old, ranging from Capitanian to Changhsingian in age (Day et al., 2015; Gastaldo et al., 2015), providing ample time for them to experience deformation during the Karoo Basin’s complex tectonic history. We predict that the severity of deformation and its effects on important biological signals will decrease on average with decreasing age of the fossils under consideration, so our results should not be used to dismiss all geometric morphometric studies of fossils out of hand. However, the existence of more recent examples such as the famously deformed Oligocene–Miocene leptaucheniine oreodonts of North America (Prothero & Sanchez, 2008) suggest a quantitative test of this hypothesis would be enlightening.

Conclusions

Distortion of Diictodon crania is not random, and can be classified into seven stereotyped deformation styles influenced by planes of compression and underlying architectural properties of the dicynodont skull. The cumulative effect of these deformation styles on morphospace, however, is indeed random dispersion around an undistorted origin. Within Diictodon feliceps, the only signal in morphological variance is associated with deformation style. Although simulated datasets indicate that it is possible to extract accurate biological signals from geometric morphomeric datasets including deformed specimens, our empirical results suggest that the more complex styles of deformation encountered when working with real specimens will likely make such success difficult in practice (mirroring the practical difficulties involved in pre-analytical retrodeformation of large samples). Therefore, we recommend caution in intraspecific analyses of variation in non-mammalian synapsids, as differences interpreted as biological variation may simply represent deformational artifact. This concern may be broadly applicable for fossil taxa, as demonstrated by other empirical and theoretical analyses (Webster & Hughes, 1999; Angielczyk & Sheets, 2007; Hedrick & Dodson, 2013; Baert, Burns & Currie, 2014; Hedrick et al., 2019). Less caution is required in the inclusion of deformed individuals in large, multispecies data sets. Although the range of morphospace occupation will be larger than in biological reality for deformed specimens, the random dispersion caused by deformation allows one to confidently infer actual position in morphospace for the within-species mean. Deformed singletons may occupy a misleading position in morphospace, but the relative import of phylogenetic signal reduces interpretive error in this case. Although deformed specimens incur minimal error of disparity metrics when within-group taxon means are used, a taxon-free approach underestimates overall disparity within the group, highlighting the need for thorough taxonomic review underlying analyses of disparity that include deformed specimens.

We thank the collections managers and curators of the many Diictodon-bearing institutions who provided access to material for this study.

Institutional Abbreviations

BP Evolutionary Studies Institute, University of the Witwatersrand, Johannesburg, South Africa

BSPG Bayerische Staatssammlung für Paläontologie und Geologie, Munich, Germany

NHMUK The Natural History Museum, London, United Kingdom

NMQR National Museum, Bloemfontein, South Africa

SAM Iziko, the South African Museum, Cape Town, South Africa

TM Ditsong National Museum of Natural History, Pretoria, South Africa

UCMP University of California Museum of Paleontology, Berkeley, USA

USNM National Museum of Natural History, Washington, D.C., USA

Additional Information and Declarations

Competing Interests

Author Contributions

Data Availability

The authors declare that they have no competing interests.

Christian F. Kammerer conceived and designed the experiments, performed the experiments, analyzed the data, prepared figures and/or tables, authored or reviewed drafts of the paper, and approved the final draft.

Michol Deutsch performed the experiments, analyzed the data, prepared figures and/or tables, authored or reviewed drafts of the paper, and approved the final draft.

Jacqueline K. Lungmus performed the experiments, analyzed the data, prepared figures and/or tables, authored or reviewed drafts of the paper, and approved the final draft.

Kenneth D. Angielczyk conceived and designed the experiments, performed the experiments, analyzed the data, prepared figures and/or tables, authored or reviewed drafts of the paper, and approved the final draft.

The following information was supplied regarding data availability:

Data and R scripts are available at Dryad Digital Repository: Kammerer, Christian; Deutsch, Michol; Lungmus, Jacqueline; Angielczyk, Kenneth (2020), Effects of taphonomic deformation on geometric morphometric analysis of fossils: a case study using the dicynodont Diictodon feliceps (Therapsida, Anomodontia), v2, Dryad, Dataset, DOI 10.5061/dryad.5tb2rbp1x.

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
