# Peer review of "Effects of taphonomic deformation on geometric morphometric analysis of fossils: a study using the dicynodont Diictodon feliceps (Therapsida, Anomodontia)"

_PeerJ, doi:10.7717/peerj.9925_

## Round 0.1 · original submission · Minor Revisions

Dear authors,

I have accepted the decision of ‘minor revisions’ from two of the three reviewers. All of the reviewers are very complimentary about your manuscript, and the review comments should be easy to address.

I look forward to receiving your revised manuscript.

·

Basic reporting

See PDF

Experimental design

See PDF

Validity of the findings

See PDF

Additional comments

Review: Kammerer et al., Effects of taphonomic deformation on geometric morphometric analysis of fossils: a case study using the dicynodont Diictodon feliceps (Therapsida, Anomodontia)

The authors aimed to examine the impacts of taphonomy on geometric morphometric analyses of fossils using Diictodon and a truly impressive sample size, not often seen in vertebrate paleontology. Using a combination of empirical and simulated datasets, they assessed the detectability of sexual dimorphism and ontogenetic trends in their Diictodon dataset. Additionally, they examined a dataset of anomodonts in addition to their Diictodon dataset to determine whether variation between higher taxonomic levels is swamped by taphonomic deformation. They found that while biologic trends were detectable in their dataset, it would be difficult to tease factors such as sexual dimorphism, ontogeny, and taphonomic variation apart in practice. Within their larger analysis, they found that phylogeny was not overprinted by taphonomy suggesting that geometric morphometric analyses on higher-level taxonomic groups will likely not be as strongly impacted by taphonomic distortion.

This is a powerful paper and a worthy addition to the literature. I have outlined several points below that would improve this paper, but I strongly feel that it should be published after minor revisions.

-Brandon Hedrick


Comments:

Line 48–51: Check the grammar in this sentence

Line 63: Note that Hedrick et al. (2019) did not use a retrodeformation technique. Rather, our paper attempted to quantify the percent of variation that could be attributed to taphonomy. It in fact has a section discussing the issues with retrodeformation. Hedrick et al. (2019) should be cited elsewhere in the paper, especially because it closely aligns with the goals you present as it also attempts to understand to what ‘degree taphonomic deformation can overwrite the underlying signal of biological shape variation.’

Line 66: Large sample sizes? Large samples sounds like ‘large specimens’

Line 102: ‘to quantify the amount…’

Line 108: Yes, this is a great point. Hedrick and Dodson (2013) discussed this at length, finding different specimens of Psittacosaurus going off in different directions depending on whether they were dorsoventrally compressed, laterally compressed, or a combination.

Line 91–120: Please add some citations here. Angielczyk and Sheets (2007), Hedrick and Dodson (2013), Tschopp et al. (2013), Baert et al. (2014), and Hedrick et al. (2019) discuss many of these ideas at length.

Line 156: add period to the end of the sentence.

Line 256: I wouldn’t say that eigenvalues to the right of the broken stick are random noise, so much as not meaningful to the overall variance structure. They usually represent singleton taxa or single landmark displacements, which are not random noise.

Line 266: How many iterations?

Line 269: DefCat citation?

Line 330: I was having a little trouble following the beginning of this paragraph. Perhaps, it could be reworded in a few places? ‘For each biological signal,…”, How many specimens were in each ‘dataset’ in each group?

Line 354: add ‘(Adams and Otárola-Castillo, 2013)’

Line 423: Interesting that anteroposterior compression is so rare. However, that is what I have qualitatively seen in my work as well. I like your interpretation as to why.

Line 487: It would be better to avoid the use of significant in colloquial terms here. Maybe substantial throughout when referring to results from the broken stick criterion?

Line 595: Was this tested with the K-mult statistic (Adams, 2014) or is this a qualitative observation?

Line 639: ’10.0’

Line 653: ‘largest’ instead of most

Line 670: ‘but not 1 through 4’?

Line 685–687: Grammar, reword

Line 739: I would be wary here. I don’t think that morphometrics should be used to directly assess taxonomy since autapomorphies axiomatically cannot be landmarked since they do not appear on all specimens. It might be worth expanding on that point here.

Line 755–765: I have wondered for a long time at what taxonomic level distortion no longer overprints biological signal and I find this point quite convincing and important.

Line 780: I like that you are up front about the difficulties of discerning biological signals in practice. However, I am not sure that looking at single taphomorphotypes would help to uncover true signals. It might be more likely to generate false signals that have higher (but spurious) correlations (e.g., if you only have laterally compressed samples and are looking for ontogenetic trends, you might get a higher correlation between shape and size than if you used multiple distortion types. However, the trends you would see in the data would be largely related to the lateral distortion). Could you expand on this idea for a few sentences?

Line 801: ‘ontogeny’

Lines 810–817: Yes, I wonder this too and speculated on it in Hedrick et al. (2019). I think future work examining taphonomic variation at different time slices will need to be done in the future to examine this question fully.

Line 834: Add Baert et al. (2014) and Hedrick et al. (2019) here

Figure 8, 10, 12: Unfortunately the colors in A and D are very difficult to discern. I would strongly suggest making these figures larger and the dots larger, especially because space is not at a premium in PeerJ. Perhaps more different colors would also help. As it is now with the resolution of figures I have, I pretty much have to take your word for where the points are. This is somewhat of a problem for figure 14–20 as well, but is helped by the additional figures with Diictodon in blue and the other groups in gray.

Not sure if you did this in R or MorphoJ, but this is a palette that I’ve found works for a large number of groups: palette(brewer.pal(12, "Paired"))

Reviewer 2 ·

Basic reporting

no comment

Experimental design

no comment

Validity of the findings

no comment

Additional comments

This is a very thorough and detailed manuscript and I can happily recommend it for acceptance with some very minor revisions. I would recommend that the authors carry out MANOVAs (for example, with the procD.lm function in geomorph) to test whether in morphology differs among groups in each of the classification schemes (deformation type, size class, sex, and assemblage zone). The interpretations of morphospace occupation are important for the conclusions of the paper and they would benefit from this type of quantitative assessment.
Line 145: Citation needed for “(well-supported sources of morphological variability in actual Diictodon)”
Line 436: why is anterodorsal shear most problematic for morphometrics?

·

Basic reporting

See general comments

Experimental design

See general comments

Validity of the findings

See general comments

Additional comments

This manuscript was a pleasure to read and I think it is basically ready for publication as-is. Interpreting taphonomic deformation and its effects on morphological disparity within and between taxa is both challenging and underappreciated by many palaeontologists, and this study provides a welcome framework for understanding the effects of taphonomic variation that I think will be very useful to vertebrate palaeontologists. The questions being addressed in this study are clearly presented, and the methods are sound (I particularly liked breaking the specimens into assemblage zones to test for taphonomic signals within the basin, in addition to size classes and sex). I have one minor suggestion that would be a helpful addition to the methods, and that would be a brief discussion about how non-taphonomically distorted specimens were identified from distorted specimens. I don’t doubt any of the classifications, but for the sake of clarity I think a few notes that you looked for things like bilateral asymmetry, elliptical deviations from circular structures like orbits or the foramen magnum, breaks/crushing/displacement, etc., would be very helpful. Thanks for the opportunity to review this extremely interesting manuscript!

---

## Round 0.2 · accepted · Accept

Dear authors,

I am pleased to inform you that I have 'accepted' your manuscript for publication.

You will shortly be contacted by the PeerJ production staff who will take you through the proof stages.

Once again, congratulations, and I hope you will continue to use PeerJ as your publication venue.

·

Basic reporting

No comment

Experimental design

No comment

Validity of the findings

No comment

Additional comments

I thank the authors for taking the majority of my comments in a previous review. I find that this paper is now ready to be accepted. It is an excellent addition to the literature. I did find a few minor grammar points that should be corrected before the proof stage, noted below.

Line 53: ‘rates’

Line 344: ‘amount’ reads weird. Maybe ‘number’?

Line 407: ‘crania’

Line 560: ‘meaningful’ rather than ‘significant’